



# Differences in Fine Particle Chemical Composition on Clear and Cloudy Days

Amy E. Christiansen[1], Annmarie G. Carlton[1], Barron H. Henderson[2]

[1]Department of Chemistry, University of California, Irvine, CA 92697, USA
[2]Office of Air Quality Planning and Standards, U.S. Environmental Protection Agency, Research Triangle Park, NC 27709, USA

*Correspondence to*: Annmarie G. Carlton (agcarlto@uci.edu)

**Abstract.** Clouds are prevalent and alter fine particulate matter ($PM_{2.5}$) mass and chemical composition. Cloud-affected satellite retrievals are subject to higher uncertainty and are often removed from data products, hindering quantitative estimates of tropospheric chemical composition during cloudy times. We examine surface $PM_{2.5}$ chemical constituent concentrations in the Interagency Monitoring of PROtected Visual Environments (IMPROVE) network in the United States during Cloudy and Clear Sky times defined using Moderate Resolution Imaging Spectroradiometer (MODIS) cloud flags from 2010-2014 with a focus on differences in particle hygroscopicity and aerosol liquid water (ALW). Cloudy and Clear Sky periods exhibit significant differences in $PM_{2.5}$ mass and chemical composition that vary regionally and seasonally. In the eastern US, relative humidity alone cannot explain differences in ALW, suggesting emissions and *in situ* chemistry exert determining impacts. An implicit clear sky bias may hinder efforts to quantitatively understand and improve representation of aerosol-cloud interactions, which remain dominant uncertainties in models.

## 1 Introduction

At any given time, visible clouds cover over 60% of the Earth's surface (King et al., 2013), and a warming climate causes cloud cover to change (Norris et al., 2016). Average cloud fraction values over the contiguous US (CONUS) are ~40% year-round with higher values in winter (44-54%) than summer (26-34%) (Ju and Roy, 2008; Kovalskyy and Roy, 2015). Clouds act as atmospheric aqueous phase reactors, and their condensed phase oxidative chemistry generates particle mass aloft, such as sulfate (Zhou et al., 2019), water-soluble organic carbon (Carlton et al., 2008; Duong et al., 2011), and organo-sulfur compounds (Pratt et al., 2013). Clouds are the primary drivers of vertical transport in the atmosphere, moving trace species from the boundary layer to the free troposphere (FT) (Ervens, 2015). The radiative impacts of aerosols in the FT are substantial, especially when located above clouds where they scatter and absorb both incoming solar radiation and diffuse back scatter from clouds (Seinfeld, 2008). Aerosol-cloud interactions are complex and a critical uncertainty in model projections (Fan et al., 2016).



Atmospheric chemistry laboratory studies, ambient sampling, modelling, and analysis strategies are often designed in ways that minimize cloud and water influences. This leads to an implicit, yet persistent clear sky bias in the quantitative understanding of tropospheric composition. During atmospheric chemistry field campaigns, aircraft typically avoid clouds, and direct measurement of in-cloud particle chemical composition is rare (Wagner et al., 2015). There is increased error in remotely sensed aerosol optical thickness (AOT) retrieval techniques during cloudy times (Martin, 2008), and impacted

retrievals are screened from final data products to avoid measurement artifacts. Most validation of satellite-derived AOT through comparison to surface measurements, such as those from sun photometers used to retrieve AOT from the ground up, is conducted for cloud-free periods (Liu et al., 2018). Air quality models are often evaluated with cloud-free satellite retrievals (van Donkelaar et al., 2010; Guo et al., 2017; de Hoogh et al., 2016; Song et al., 2014; Tian and Chen, 2010) and cloud-free aircraft samples (Bray et al., 2017; McKeen et al., 2009). This biases model development and predictive skill

toward cloud-free conditions, and hinders accurate prediction of trace species during cloudy time periods. Laboratory experiments to understand particulate matter formation are conducted under dry conditions (Lamkaddam et al., 2017; Ng et al., 2007) atypical of cloudy time periods. Should differences in aerosol physicochemical properties, including those that affect water uptake, exist between cloudy and clear sky time periods, current approaches are limited in their ability to quantitatively assess those differences. This is a key knowledge gap.


      Characterization of fine particulate matter ($PM_{2.5}$) mass and chemical composition in the US primarily relies on surface measurements from relatively sparsely spaced monitors. At various locations across the CONUS, the Interagency Monitoring of PROtected Visual Environments (IMPROVE) network samples every 3 days, and the Chemical Speciation Network (CSN) samples every 3 or 6 days (US Environmental Protection Agency, 2008). To improve upon surface network spatial

and temporal limitations, data can be interpolated to describe particle mass (Li et al., 2014; Zhang et al., 2018) and chemical composition over larger areas (Liu et al., 2009; Tai et al., 2010). Satellite information can also be used (van Donkelaar et al., 2015b), such as the Moderate Resolution Imaging Spectroradiometer (MODIS) instruments aboard the Aqua and Terra satellite platforms. These view the entire Earth surface every 1 to 2 days and are used to impart information for use in air quality applications (van Donkelaar et al., 2015b; Gupta et al., 2006; Kloog et al., 2011; Sorek-Hamer et al., 2016). Many

advanced satellite AOT models translate space-based radiation measurements to surface $PM_{2.5}$ (van Donkelaar et al., 2010, 2015b, 2015a; Gupta et al., 2006; Kessner et al., 2013; Kloog et al., 2011; Kumar et al., 2007; Liu et al., 2011; Schaap et al., 2009; Wang et al., 2012; Wang and Christopher, 2003) and employ sophisticated techniques which account for aerosol size and type, vertical extinction, mass, and relative humidity (RH) (van Donkelaar et al., 2010). Evaluation of AOT-to-$PM_{2.5}$ techniques finds that monthly aggregated AOT can robustly estimate relationships spanning five years of daily mean values

over North America (R>0.77) (van Donkelaar et al., 2010). While temporal and geospatial satellite AOT is useful for understanding trends in $PM_{2.5}$ concentrations (van Donkelaar et al., 2015b; Sorek-Hamer et al., 2016; Wang and Christopher, 2003), an implicit constraint for this and other similar findings is that such agreement is for clear sky conditions.



Surface networks record PM$_{2.5}$ mass and chemical composition during clear sky and cloudy time periods alike. The
difference between spatially and temporally aggregated PM$_{2.5}$ mass concentrations in the CONUS for cloudy and all sky
(cloudy + clear sky) conditions is estimated to be ±2.5 μg m$^{-3}$ (Christopher and Gupta, 2010). Less attention has been given
to clear sky and cloudy differences in PM$_{2.5}$ chemical composition, especially with regards to particle hygroscopicity and
water uptake. Aerosol mass concentrations and chemical speciation including aerosol liquid water (ALW) influence AOT
(Christiansen et al., 2019; Malm et al., 1994; Nguyen et al., 2016; Pitchford et al., 2007), cloud microphysics, and mesoscale
convective systems (Kawecki and Steiner, 2018), including storm morphology and precipitation patterns (Kawecki et al.,
2016). Particle chemical composition modulates particle size via water uptake. Particle size is a determining factor in light
scattering by particles, which is important for aerosol radiative calculations. An implication of this work is that if particle
hygroscopicity changes from clear sky to cloudy time periods, when aerosol-cloud interactions are most important, a
quantitative understanding remains unclear.

In this work, we test the hypothesis that there are quantitative differences in PM$_{2.5}$ chemical composition between cloudy and
clear sky time periods in ways important for water uptake. We employ a combination of satellite products, surface
measurements, and thermodynamic modeling to analyze annual and seasonal trends in chemical climatology regions across
the CONUS. We assess and quantify seasonal statistical significance (Kahn, 2005) for differences in distributions of RH,
PM$_{2.5}$, and chemical speciation during cloudy and clear sky times using surface measurements from the IMPROVE network
from 2010-2014 within the context of MODIS cloud flag values. Further, we examine one chemical climatology region in
detail, the Mid South, as a case study. This region encompasses the location of the Atmospheric Radiation Measurement
Southern Great Plains (SGP) site in an area of the CONUS that experiences varied weather patterns, a broad range of cloud
conditions, and distinct seasonal variations in temperature and humidity (Sisterson et al., 2016).

**2 Data and Methods**

Cloudy and clear sky classifications are determined using publicly available data (National Aeronautics and Space
Administration, 2018) from MODIS on the Aqua and Terra satellites. Pairing of satellite and surface PM$_{2.5}$ mass
measurements typically works best in rural and vegetated locations, where the spectral properties of the background tend to
be dark and vary little over the space of a satellite grid cell (Hauser, 2005; Jones and Christopher, 2010). For this reason, we
use rural IMPROVE network sites that are located primarily in national parks, although improvements have been made for
retrievals over bright surfaces (Hauser, 2005; Hsu et al., 2004, 2006, 2013; Zhang et al., 2016). We use 500 m resolution
pixels that contain the IMPROVE sites. Retrievals are flagged as cloudy if QA flags specifically identified clouds as
preventing retrieval, or if 2.1-micrometer reflectance was too high (r>0.35) and the fraction of 500 m sub pixels that were
cloudy was greater than 44.4%. We choose 44.4% because it is a fundamental limit of the algorithm (Remer et al., 2013).
IMPROVE monitors are frequently under a MODIS swath with valid retrievals even if the pixel containing the IMPROVE



station is not successfully retrieved. As an alternative to the IMPROVE pixel, we employ a method for quality assurance, a 17x17 grid. This allows for any retrieval within a 50 km x 50 km area to represent the IMPROVE station. If all 17x17 pixels are not retrieved, then the state over the monitor is determined to be cloudy. The 17x17 grid approach is much more likely to attribute non-retrieved data to clouds (98.5%) than the containing pixel approach, which attributes 89.8% of non-retrieved

data to clouds. Misidentifying non-retrievals as cloudy is unlikely to substantially affect interpretation, as the sample size is large (N>70,000 total observations, and N>1500 for an individual region).

IMPROVE network data were downloaded on 13 July 2015 and 26 May 2016 from public archives (http://vista.cira.colostate.edu/Improve/) (IMPROVE Network, 2019) for 132 unique sites across the CONUS with complete

data records for the years 2010-2014 (Fig. S1a). IMPROVE data is collected every 3 days. We investigate 24-hour average $PM_{2.5}$ mass, ALW, RH, sulfate ($SO_4^{2-}$), nitrate ($NO_3^-$), and total organic carbon (TOC) mass concentrations. Other species affect particle hygroscopic properties but are not widely measured in routine networks. For example, we investigate TOC as a whole even though primary and secondary species affect water uptake differently. There is no direct measurement of either in routine monitoring network operations, although fractionation can sometimes be used to infer information about sources

and formation processes (Aswini et al., 2019; Cao et al., 2005; Chow et al., 2004). We group IMPROVE sites across the CONUS into 22 chemical climatology regions defined by the IMPROVE network (Fig. S1b) (Hand et al., 2011; Malm et al., 2017). $PM_{2.5}$ mass and composition is provided directly from the IMPROVE database, while ALW is estimated.

ALW is a function of RH, particle concentration, and chemical composition. We estimate ALW using a metastable

assumption in the inorganic ($K^+$–$Ca^{2+}$–$Mg^{2+}$–$NH_4^+$–$Na^+$–$SO_4^{2-}$–$NO_3^-$–$Cl^-$–$H_2O$) aerosol thermodynamic equilibrium model ISORROPIAv2.1 (Fountoukis and Nenes, 2007). We use the reverse, open-system problem because only aerosol measurements are available. Particle mass concentration inputs of $SO_4^{2-}$ and $NO_3^-$ are taken from IMPROVE measurements. Ammonium ion is not considered due to limited measurement availability. Dust and organic species are also not considered because water uptake properties are not well constrained (Jathar et al., 2016; Metzger et al., 2018), and there is large spatial

heterogeneity in dust. Our approach to employing ISORROPIA introduces uncertainties (e.g., pH estimates would be unreliable (Guo et al., 2015)), but neglect of dust does not affect overall interpretation of ALW mass (Fig. S2), consistent with an earlier sensitivity using this technique that included organic species (Nguyen et al., 2015). The temperature and RH were extracted from the North American Regional Reanalysis (NARR) model (Kalnay et al., 1996) similar to Nguyen et al. 2016.


Cloudy and Clear Sky differences in ALW are investigated in two ways. First, we compare ALW estimated using 24-hour average chemical composition and meteorology and group results into Clear Sky and Cloudy bins using the MODIS cloud flag. We use these daily values when comparing ALW within chemical climatology regions. Second, we investigate trends across the eastern US to isolate the effect of chemical composition. We select the eastern US since ALW concentrations are





largest in this region (Fig. S3), and it is in cloud often and consistently (cloud fraction 30-50% year-round) (Fig. S4). This makes statistical comparisons between Cloudy and Clear Sky times more robust than in the dry western US, where ALW concentrations and cloud fraction are low in most seasons. We group 24-hour average chemical composition and meteorology into Clear Sky and Cloudy bins and take monthly medians. We perform ALW estimations using the medians via three ISORROPIA calculation scenarios: 1) Clear Sky chemical composition and Clear Sky meteorology ("Clear Sky"

scenario), 2) Cloudy chemical composition and Cloudy meteorology ("Cloudy"), and 3) Clear Sky chemical composition and Cloudy meteorology ("Mixed") (Table S1, Fig. S5). We use monthly medians to avoid complications that arise from differing numbers of Cloudy and Clear Sky days in the Mixed scenario. To investigate meteorology and chemical composition impacts separately, we perform the Mixed scenario in order to reproduce studies in which cloud free growth factors (Brock et al., 2016) are eventually applied in models that contain cloudy meteorological conditions (Bar-Or et al.,

2012). When the Mixed scenario is significantly different than Cloudy, we can reject the hypothesis that RH and temperature alone explain the difference. Wet deposition is unconstrained in this analysis, but cloud droplets typically evaporate (Pruppacher and Klett, 2010).

Growth factors used in the Mid South region are estimated from a modified Kohler equation (Brock et al., 2016; Jefferson et

al., 2017) (Eq. 1). We use RH from the NARR and estimate $\kappa_d$, the particle hygroscopicity, from IMPROVE-measured chemical composition mass concentrations and individual species κ values ($\kappa_{SO4}$=0.5, $\kappa_{NO3}$=0.7) (Petters and Kreidenweis, 2007). Here, $gf(D)$ is the hygroscopic diameter growth.

$$gf(D) = (1 + \kappa_d \frac{RH}{100-RH})^{1/3}$$   (1)


Statistical significance for differences in measurement distributions of PM$_{2.5}$ chemical composition and properties between Cloudy and Clear Sky time periods from 2010-2014 is determined using the Mann-Whitney U Test in R statistical software (R Core Team, 2013). The Mann-Whitney U Test is a non-parametric test that compares two samples to assess whether population distributions differ (McKnight and Najab, 2010). 2010-2014 encompasses typical conditions and coincides with

several intensive observation periods including the Southeast Atmosphere Studies (SAS) (Carlton et al., 2018), the Studies of the Emissions and Atmospheric Composition, Clouds, and Climate Coupling by Regional Surveys (SEAC[4]RS) (Toon et al., 2016), and the California Research at the Nexus of Air Quality and Climate Change (CalNex) (Ryerson et al., 2013) field campaigns. We define cloud fraction for each region as the number of MODIS-flagged cloudy IMPROVE sampling days over the total number of IMPROVE sampling days. Further, we define winter as December, January, and February (DJF),

spring as March, April, and May (MAM), summer as June, July, and August (JJA), and fall as September, October, and November (SON).



## 3 Results and Discussion

### 3.1 Hygroscopicity and Chemical Composition

Distributions in monthly particle chemical composition across the eastern US in 2010-2014 are sufficiently changed between

MODIS-defined Cloudy and Clear Sky times to affect hygroscopicity and alter predicted ALW mass concentrations beyond differences that would arise from changes in meteorology alone (Fig. 1). These findings are consistent with an analysis in the desert southwest US that shows that chemical composition is an essential factor for improving cloud condensation nuclei predictions (Crosbie et al., 2015). The only difference between the Mixed and Cloudy ALW calculations is that the Mixed scenario employs Clear Sky chemical composition (rather than Cloudy chemical composition) extrapolated to Cloudy

meteorology. This type of scenario can occur in model development or satellite validation applications when $PM_{2.5}$-AOD relationships or growth factors remain unmeasured for Cloudy periods (Brock et al., 2016; van Donkelaar et al., 2010; de Hoogh et al., 2016; Tian and Chen, 2010). Previous work using climate models shows that application of ALW uptake that is influenced by incorrect chemical composition significantly affects top of atmosphere radiative forcing estimates and attribution of anthropogenic climate impacts (Rastak et al., 2017). When Clear Sky chemical composition is extrapolated to

Cloudy period meteorology ("Mixed"), monthly median ALW concentrations in the eastern US, in all seasons except winter, are significantly different from our best estimate, which employs the actual chemical composition during cloudy periods ("Cloudy"). Interestingly, monthly median Clear Sky and Cloudy scenario ALW concentrations do not differ significantly except during winter despite higher Cloudy RH (Fig. 2). This is consistent with chemical composition as a determining factor in ALW (Carlton and Turpin, 2013; Liao and Seinfeld, 2005), CCN (Crosbie et al., 2015), and extinction (Pitchford et

al., 2007) on cloudy days because the pattern in ALW is opposite the pattern in RH.

Clear Sky/Cloudy patterns in $SO_4^{2-}$ and $NO_3^-$ mass concentrations, which affect particle hygroscopicity, vary regionally and seasonally. When aggregated over the eastern US, ALW estimates for the Mixed case are largest during summer and spring and can be explained by elevated Clear Sky $SO_4^{2-}$ and $NO_3^-$ concentrations and high Cloudy RH (Fig. 2). Generally, Mixed

ALW concentrations in the eastern US are higher than for the Cloudy scenario because Clear Sky chemical composition facilitates greater hygroscopicity and Cloudy RH is elevated (Table S2). A notable exception is the Ohio River Valley during winter, where Cloudy $SO_4^{2-}$, $NO_3^-$, and RH are higher than Clear Sky. In this case, Cloudy period ALW concentrations are higher than for the Mixed scenario. These findings highlight that a changing $PM_{2.5}$ chemical composition has a determining effect on ALW mass concentrations (Nguyen et al., 2016), a critical element in the estimation of aerosol-cloud interactions

and particle radiative impacts. During cloudy periods, when the accurate prediction of ALW and aerosol-cloud interactions is most critical, *in situ* knowledge of $PM_{2.5}$ chemical composition is required.

Differences in daily mass concentrations of fine particle chemical constituents between Cloudy and Clear Sky periods across the CONUS are spatially and temporally different among $PM_{2.5}$ mass and its chemical constituents except in the Northwest





region (Figs. 3-7, Tables S3-S7, Fig. S6). These patterns cannot be adequately described as a function of MODIS cloud fraction (Figs. S6-S7). If meteorological processes and physical transport are the only controlling factors, then patterns in mass concentrations among $PM_{2.5}$ and constituents should not vary. However, they do, suggesting differences in emissions and/or *in situ* chemical production of $PM_{2.5}$ during Cloudy and Clear Sky time periods. Where differences are significant for ALW, Cloudy ALW is higher than Clear Sky in all seasons, with few exceptions (Fig. 3, Table S3). Water uptake

contributes to particle growth with a determining impact on particle size and radiative properties. $PM_{2.5}$ mass, greater during Clear Sky times in most regions and seasons, has nearly an opposite pattern to ALW spatial and seasonal trends (Fig. 4). The largest ALW differences are observed in the central and eastern US during winter. Wintertime Cloudy $SO_4^{2-}$ mass concentrations are greater than Clear Sky (Fig. 5, Table S5), and the highest $NO_3^-$ mass concentration differences are observed during Cloudy times in winter when temperatures are coldest (Fig. 6, Table S8). This promotes thermodynamic

stability of nitrate in the condensed phase, increasing particle hygroscopicity and facilitating ALW.

Outside of winter, significant $SO_4^{2-}$ mass concentrations are typically higher on Clear Sky days in the eastern US (Fig. 5, Table S5). Higher Clear Sky $SO_4^{2-}$ concentrations during summertime are associated with heat waves and stagnation events, which are characterized by a lack of ventilation in high pressure systems (Jacob and Winner, 2009; Wang and Angell, 1999)

and higher electricity demand (Farkas et al., 2016) associated with emissions that form sulfate.

TOC mass concentrations are nearly always higher during Clear Sky times than Cloudy (Fig. 7, Table S7) in all chemical climatology regions across the CONUS, with the largest differences during summer and fall. Precursor VOC emissions (e.g., biogenic) and subsequent derived PM that contributes to OC differ by season and region (Donahue et al., 2009; Gentner et

al., 2017; Youn et al., 2013). Increased sunlight under clear sky conditions leads to higher biogenic VOC emissions (Sakulyanontvittaya et al., 2008) and enhanced photolysis rates that facilitate hydroxyl radical production important to secondary organic aerosol formation (Tang et al., 2003). Organic aerosol hygroscopicity and water uptake is highly uncertain (Christiansen et al., 2019; Nguyen et al., 2015), and yet has profound impacts on top-of-atmosphere radiative forcing calculations (Rastak et al., 2017). We note that TOC is also influenced by primary sources of OC including wildland fires in

the west and prescribed burning in the east which are not influenced by cloud presence (Spracklen et al., 2007; Tian et al., 2009; Zeng et al., 2008).

**3.2 $PM_{2.5}$ Mass Concentrations**

Significant differences in $PM_{2.5}$ mass concentrations measured at IMPROVE monitoring locations are observed between Cloudy and Clear Sky conditions in the majority (>60%) of regions in any given season during 2010-2014 (Fig. 4 and Table

S4) and do not correlate with MODIS cloud fraction during any season in any region (Fig. S8). In all regions, Clear Sky $PM_{2.5}$ concentrations are generally higher than Cloudy. Satellite AOT products used to derive $PM_{2.5}$ may overestimate the atmospheric burden across the CONUS, particularly during summertime. Median All Sky $PM_{2.5}$ concentrations are also





significantly different and typically lower than Clear Sky in multiple chemical climatology regions (Table S9). This suggests the clear sky bias in satellite data may impart a positive bias when assessing surface $PM_{2.5}$ trends in model applications for

air quality, weather, and climate.

### 3.3 Case Study: The Mid South

ALW concentrations are significantly higher during Cloudy times than Clear Sky in the Mid South during all seasons (Table 1, Fig. 8). RH in the region is high year-round during Cloudy and Clear Sky periods alike, with the median greater than 60%. Gas-phase water vapor mixing ratios are sufficiently high that water availability is not limiting for ALW in the region for

any season. Aerosol mass concentrations and chemical composition vary, however, and the effects on particle hygroscopicity can be seen in contrasting Cloudy and Clear Sky ALW concentrations among the seasons. For example, during Clear Sky conditions, the highest ALW mass concentrations occur during summer and spring, which correspond to the highest $SO_4^{2-}$ concentrations in the Mid South, and not when Clear Sky RH is highest (i.e., during winter). The largest absolute ALW concentrations and estimated growth factors occur during Cloudy times in the winter and spring, when $NO_3^-$ mass fraction

and RH are highest. This is consistent with independent humidified nephelometer measurements by Jefferson et al., who find that aerosol growth rates are highest in the winter and spring at the SGP site within the Mid South chemical climatology region, and identify nitrate and RH as determining factors (Jefferson et al., 2017).

$NO_3^-$ concentrations are generally lower than $SO_4^{2-}$ in the Mid South, but $NO_3^-$ is more hygroscopic and provides influence

over ALW patterns. Sulfate is traditionally considered dominant in determining absolute ALW mass concentrations in this region, and sulfate mass fraction is highest in summer (Carlton and Turpin, 2013; Gasparini et al., 2006). Similar to other regions of the CONUS, $SO_4^{2-}$ mass concentrations are greatest during summertime Clear Sky conditions due to transport (Parworth et al., 2015), increased rates of photochemistry (Stone et al., 2012), and increased electricity sector emissions during heat waves and stagnation events (Appel et al., 2011; Farkas et al., 2016), which generally occur on sunny days.

Sulfate mass fraction is lowest in winter, when $NO_3^-$ concentrations are high due to cooler temperatures and transport of precursor species from nearby agricultural and surrounding urban areas (Parworth et al., 2015). Year-round $NO_3^-$ concentrations are higher during Cloudy conditions than Clear Sky, which are associated with lower temperatures. Under Cloudy conditions, the highest ALW concentrations and estimated growth factors occur during winter and spring, when $NO_3^-$ mass fraction and RH are highest. In another continental location, the Po Valley in Italy, $NO_3^-$ was found to control ALW

concentrations with implications for secondary organic aerosol (Hodas et al., 2014). The Mid South is also a continental, agricultural area and aerosol growth may be subject to similar mechanisms.



## 4 Conclusions

Across the CONUS, statistically discernible differences among $PM_{2.5}$ and chemical constituent concentrations under Cloudy and Clear Sky conditions cannot be explained solely by physical mechanisms. The chemical properties of aerosol are important to explain differences in water uptake and particle composition under different meteorological conditions. While meteorological phenomena such as pressure systems, winds, and air mixing affect $PM_{2.5}$ and chemical component concentrations, they are insufficient to explain chemical constituent differences between Cloudy and Clear Sky times. *In situ* chemical formation processes are necessary to fully explain temporal and spatial patterns. Spatially and seasonally, $PM_{2.5}$ and particle speciation information that lends insight into water uptake, particle properties, and particle growth is incomplete when information is gathered only during Clear Sky times. The work presented here indicates aerosol growth due to water uptake is greatest during satellite periods identified as Cloudy in many regions, when satellites are unable to remotely sense particle properties and impacts. This limits understanding of atmospheric particle burden and its climate-relevant physicochemical properties, which have implications for the prediction of weather (Kawecki and Steiner, 2018), air quality, and climate. This indicates that the clear sky bias affects accurate representation of ALW on cloudy days and suggests that without *in situ* chemical information, aerosol-cloud interactions and subsequent estimates of radiative forcings in models (Lin et al., 2016; Vogelmann et al., 2012) will remain a large uncertainty.

## Acknowledgements

This research was funded, in part, by NSF Grant AGS-1242155 and NASA Grant #80NSSC19K0987. The views expressed in this manuscript are those of the authors and do not necessarily reflect the views or policies of the U.S. Environmental Protection Agency. The IMPROVE database can be found at http://vista.cira.colostate.edu/Improve/. NCEP Reanalysis data are available from the NOAA/OAR/ESRL PSD in Boulder, Colorado, United States, at http://www.esrl.noaa.gov/psd/. MODIS data were acquired from the NASA Global Change Master Directory (GCMD) at https://gcmd.nasa.gov/. The authors thank Virendra Ghate for support in retrieving NCEP Reanalysis data, and Divya Srivistava and Julia Daniels for technical support. The authors also thank Athanasios Nenes for the development and public availability of ISORROPIA.

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






**Table 1: Particle chemical constituent concentrations, meteorology, and growth factors during Cloudy (Cl) and Clear Sky (CS) times in the Mid South.**

| | $SO_4^{2-}$ | | $NO_3^-$ | | ALW | | RH | | Growth Factors | |
|---|---|---|---|---|---|---|---|---|---|---|
| | CS | Cl | CS | Cl | CS | Cl | CS | Cl | CS | Cl |
| **Win** | 0.77 | 1.24 | 0.90 | 1.22 | 1.32 | 3.61 | 0.64 | 0.80 | 1.33 | 1.50 |
| **Spr** | 1.46 | 1.79 | 0.37 | 0.50 | 2.48 | 4.02 | 0.62 | 0.76 | 1.25 | 1.41 |
| **Sum** | 1.91 | 1.69 | 0.20 | 0.19 | 2.92 | 3.57 | 0.59 | 0.72 | 1.21 | 1.39 |
| **Fall** | 1.05 | 1.17 | 0.18 | 0.33 | 1.56 | 2.74 | 0.57 | 0.73 | 1.18 | 1.37 |







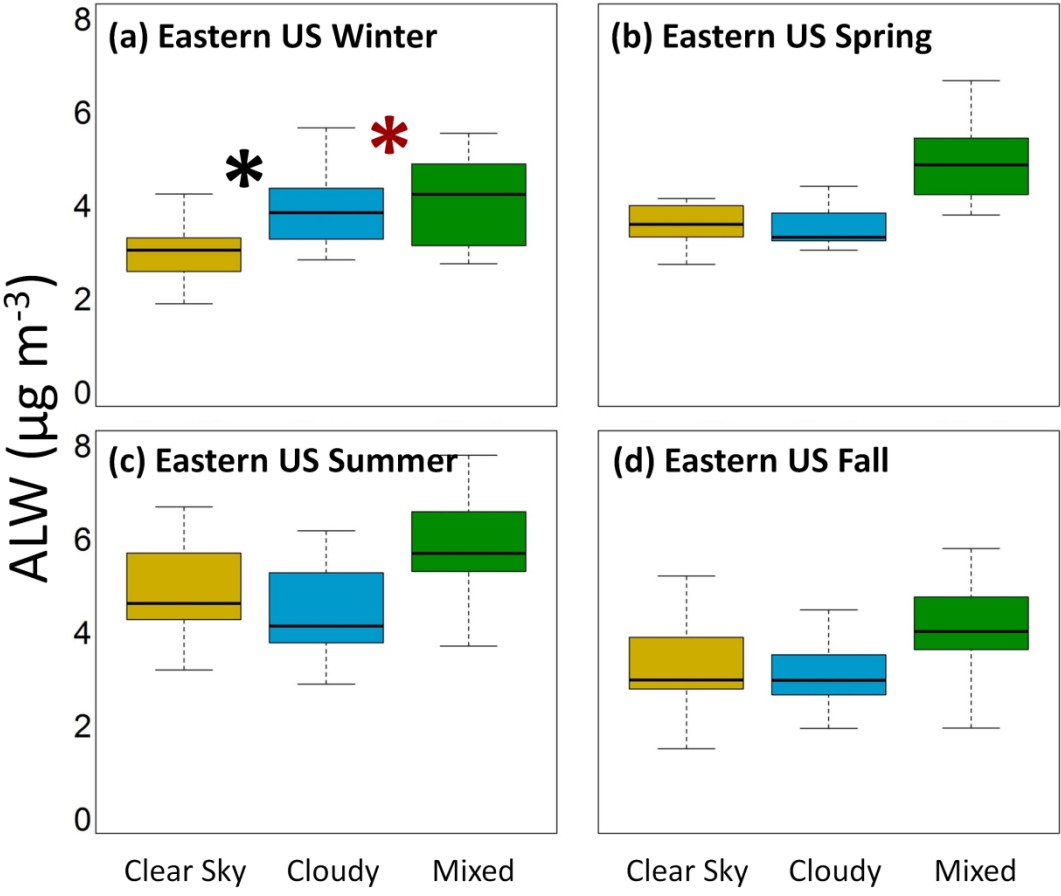

**Figure 1: ALW mass concentrations are significantly different between Clear Sky and Cloudy time periods beyond what would arise from changes solely in meteorology (e.g., RH). Monthly median estimated ALW distributions at each IMPROVE monitor in the eastern US during Clear Sky times (yellow, Clear Sky scenario), Cloudy times (blue, Cloudy scenario), and Cloudy times employing Clear Sky particle chemical composition (green, Mixed scenario). The black asterisk in (a) indicates the only situation where Clear Sky and Cloudy scenarios differ significantly. The red asterisk in (a) indicates the only situation where the Cloudy and Mixed scenarios do not differ significantly. The midline in the box is the median, the box boundaries are the 25th and 75th percentiles, and the whiskers are the 10th and 90th percentiles. Potential outliers are not shown but are used in calculations.**





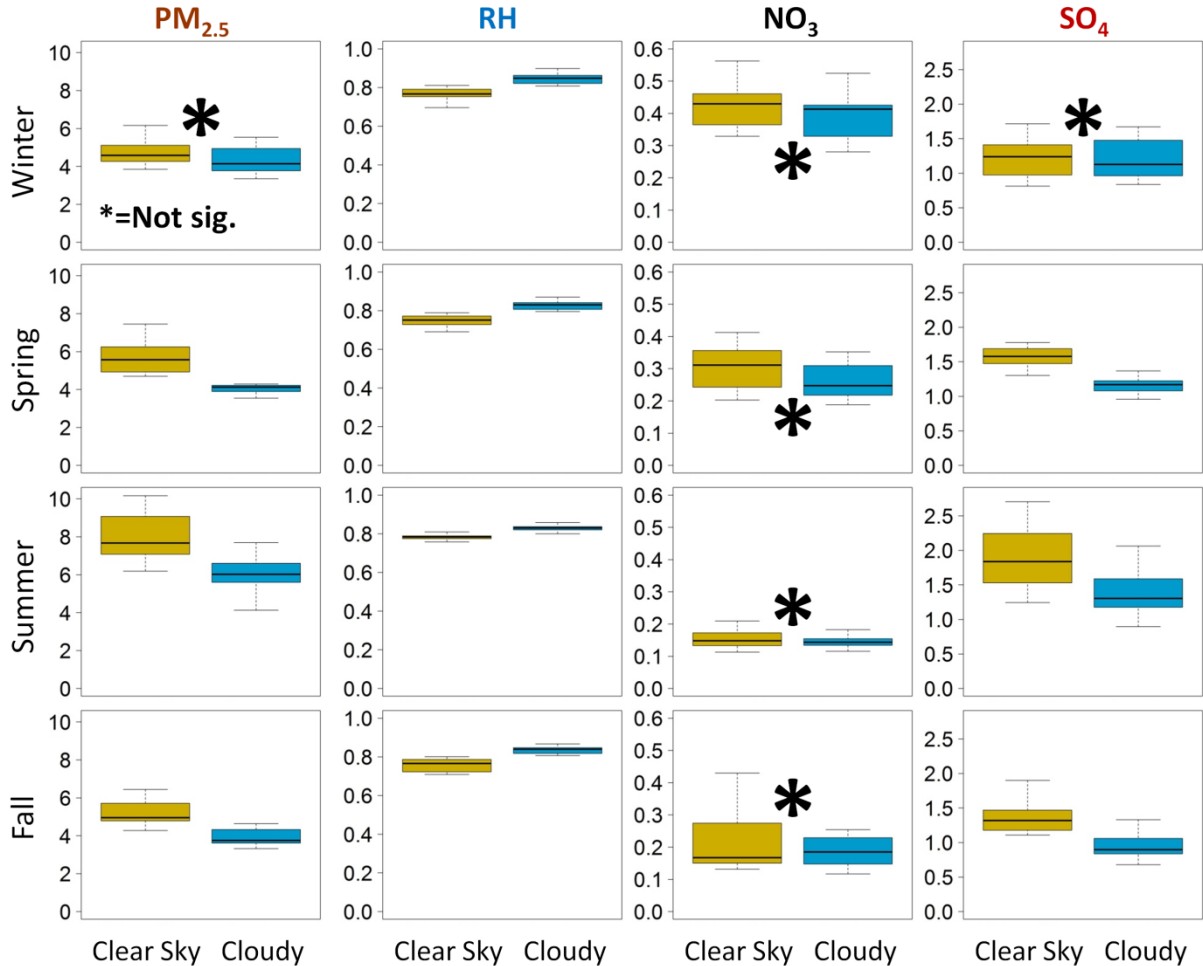

**Figure 2: Box plots of PM$_{2.5}$, RH, NO$_3^-$, and SO$_4^{2-}$ during clear sky times (yellow) and cloudy times (blue) across the eastern US.**
**Note that potential outliers are not shown but are used in calculations. The width of the box plot is proportional to the number of**
**observations. Asterisks denote Cloudy and Clear Sky differences that are *not* significant (p<0.05) by the Mann-Whitney U Test.**


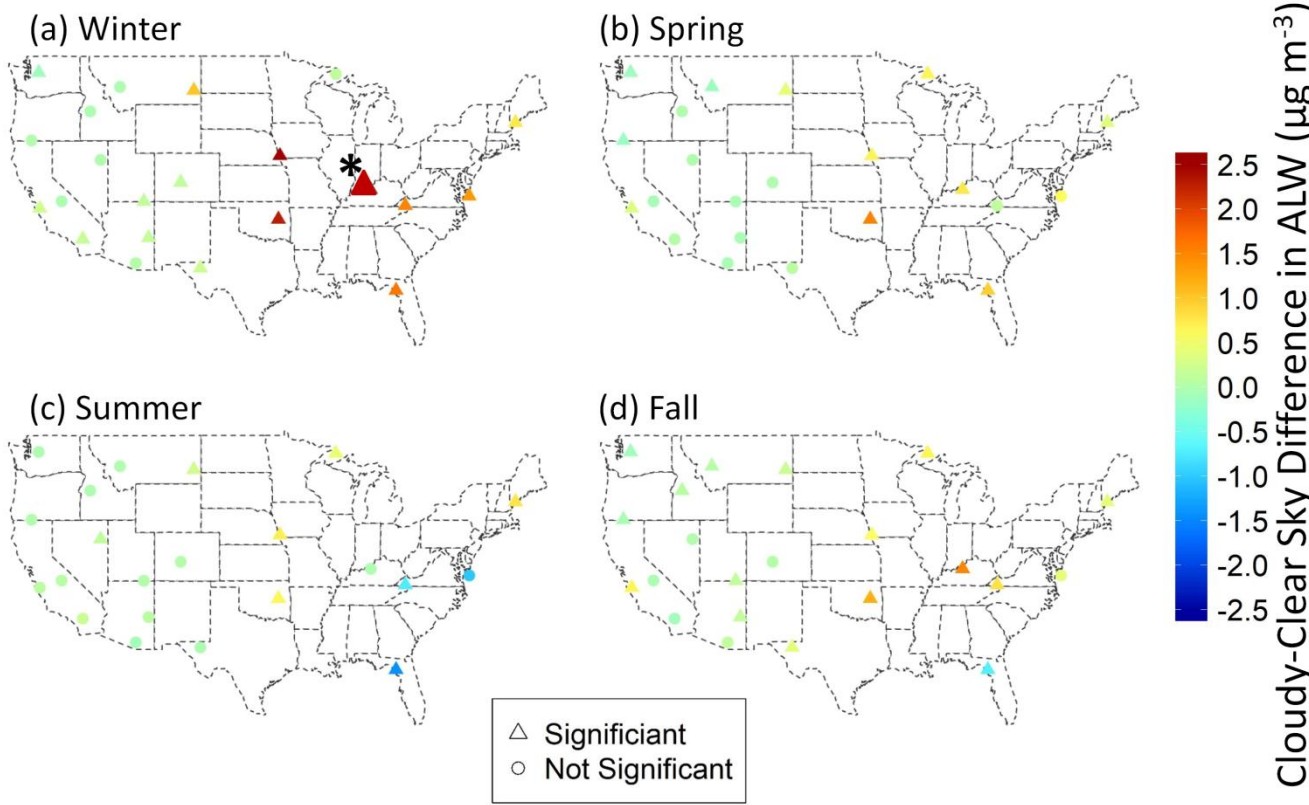

**Figure 3: Maps of the difference in ALW mass concentration medians (Cloudy-Clear Sky) for all regions from 2010-2014 for (a) winter, (b) spring, (c) summer, and (d) fall. The color of the point corresponds to the magnitude of the difference. Triangles indicate that median differences are significant by the Mann-Whitney U Test. Note that the difference in wintertime medians for daily ALW concentrations in the Ohio River Valley (denoted with asterisk) is substantially larger than other regions (Cloudy median value is 4.6 µg m$^{-3}$ larger than Clear Sky).**



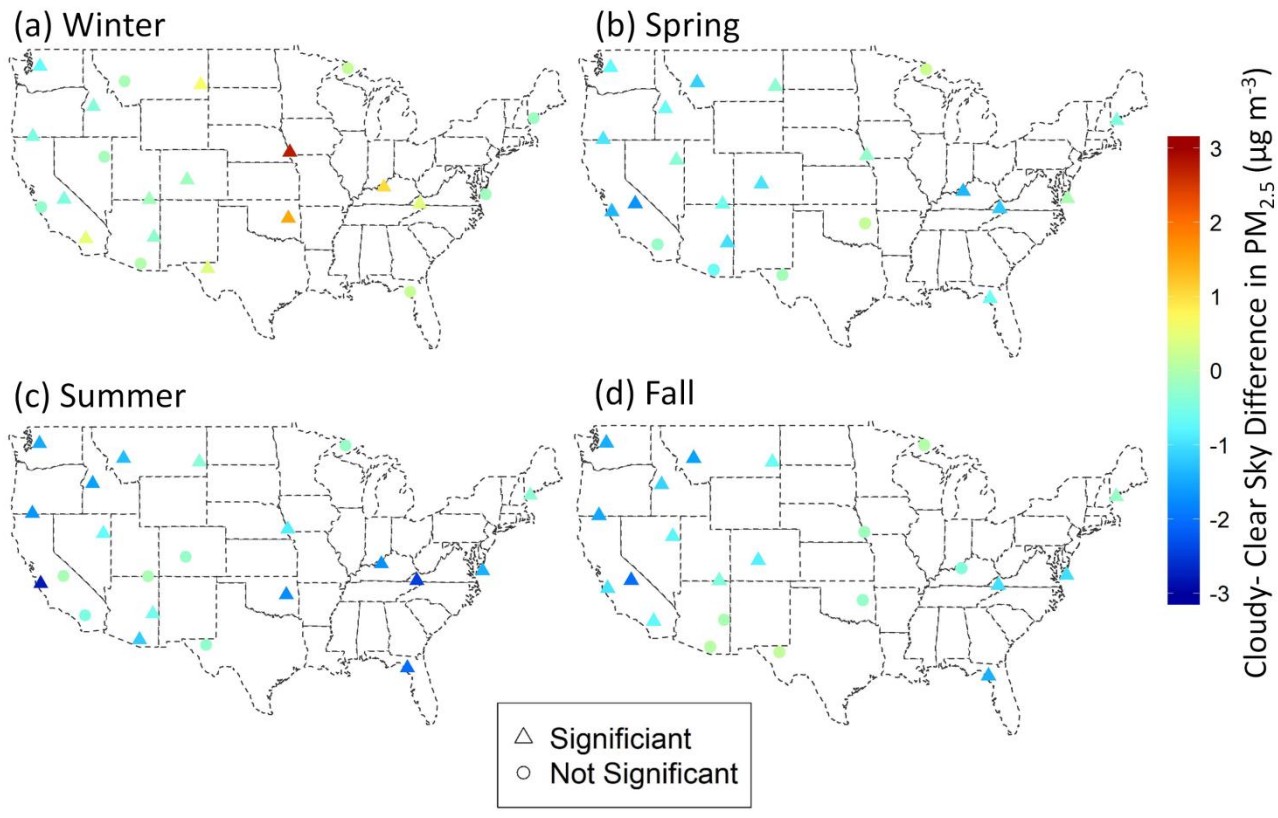


**Figure 4: Maps of the difference in PM$_{2.5}$ mass concentration medians (Cloudy-Clear Sky) for all regions from 2010-2014 for (a) winter, (b) spring, (c) summer, and (d) fall. The color of the point corresponds to the magnitude of the difference. Triangles indicate that median differences are significant by the Mann-Whitney U Test.**

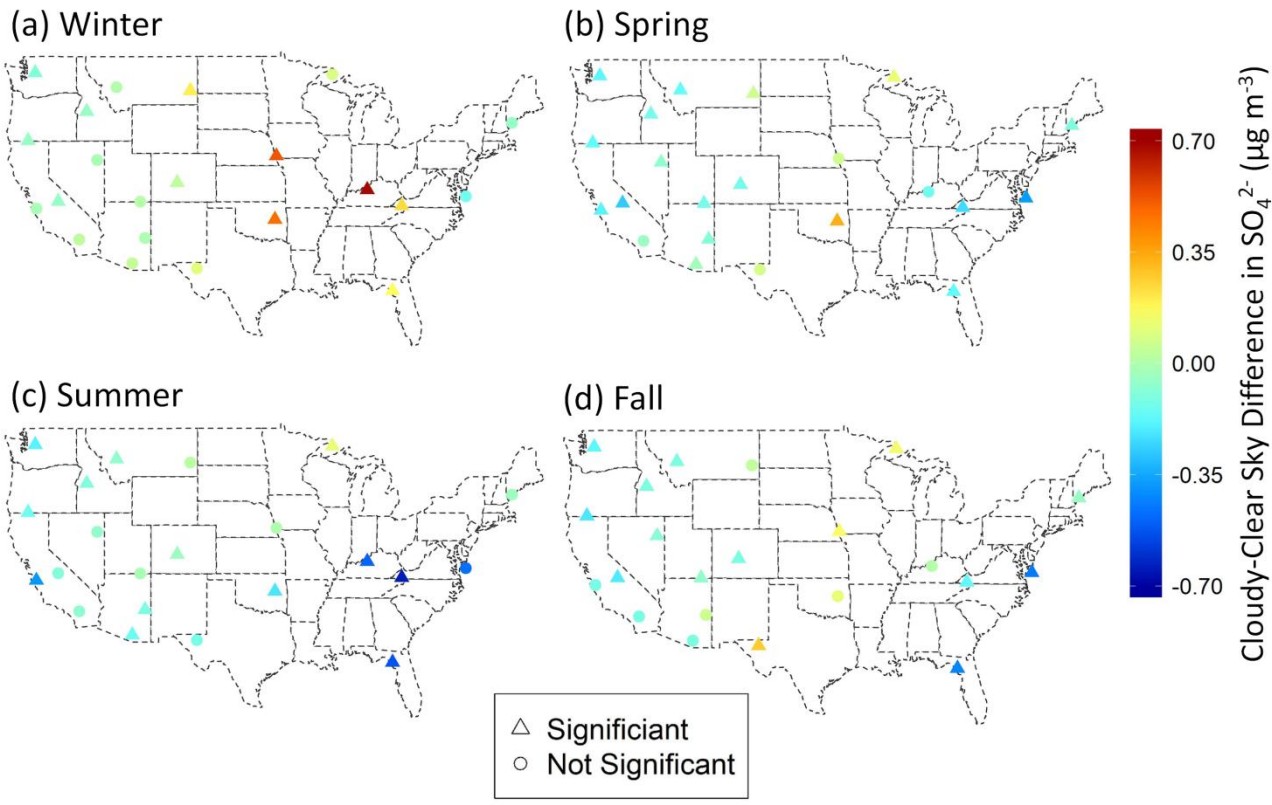


**Figure 5: Maps of the difference in SO$_4^{2-}$ mass concentration medians (Cloudy-Clear Sky) for all regions from 2010-2014 for (a) winter, (b) spring, (c) summer, and (d) fall. The color of the point corresponds to the magnitude of the difference. Triangles indicate that median differences are significant by the Mann-Whitney U Test.**




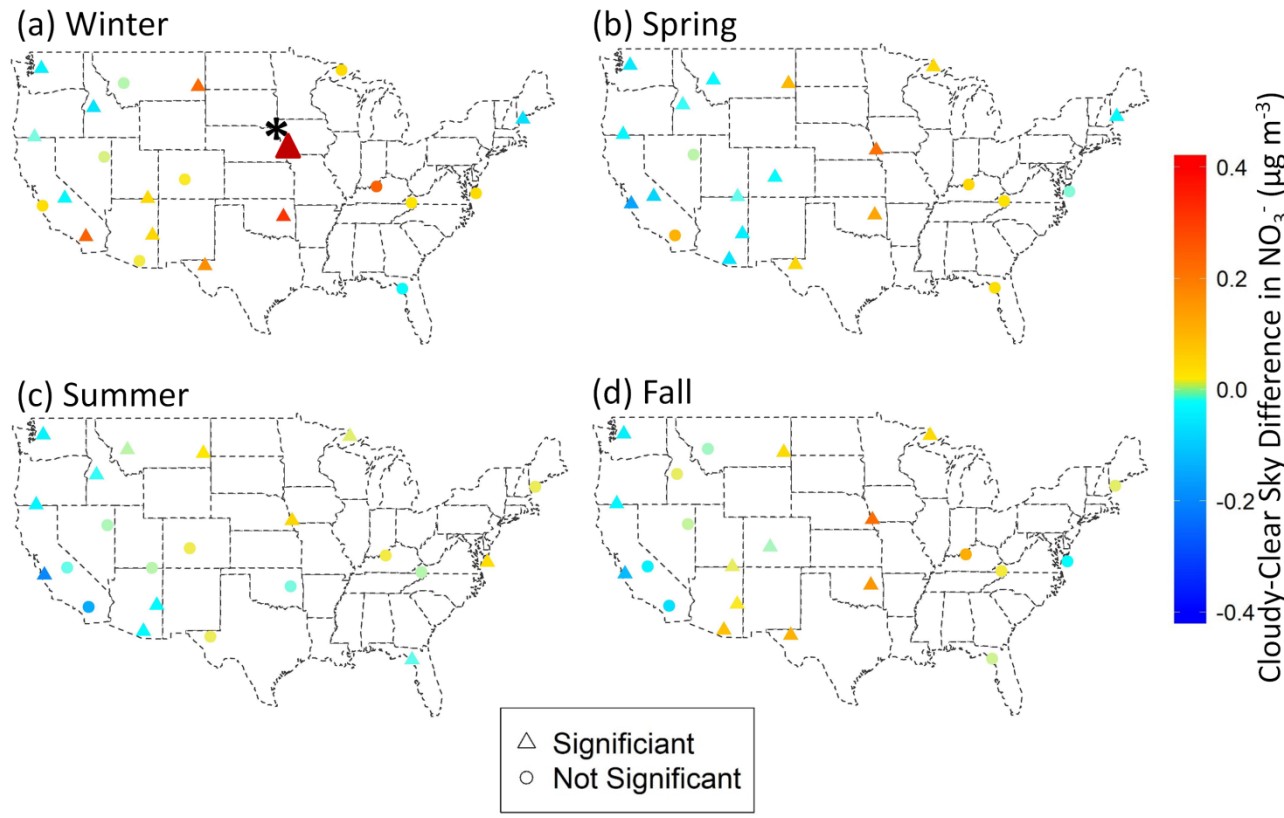

Figure 6: Maps of the difference in NO$_3^-$ mass concentration medians (Cloudy-Clear Sky) for all regions from 2010-2014 for (a) winter, (b) spring, (c) summer, and (d) fall. The color of the point corresponds to the magnitude of the difference. Triangles indicate that median differences are significant by the Mann-Whitney U Test. Note that the difference in medians for daily NO$_3^-$ concentrations in winter for the Central Great Plains (denoted with asterisk) is substantially larger than other regions (Cloudy median value is 1.07 µg m$^{-3}$ larger than Clear Sky).

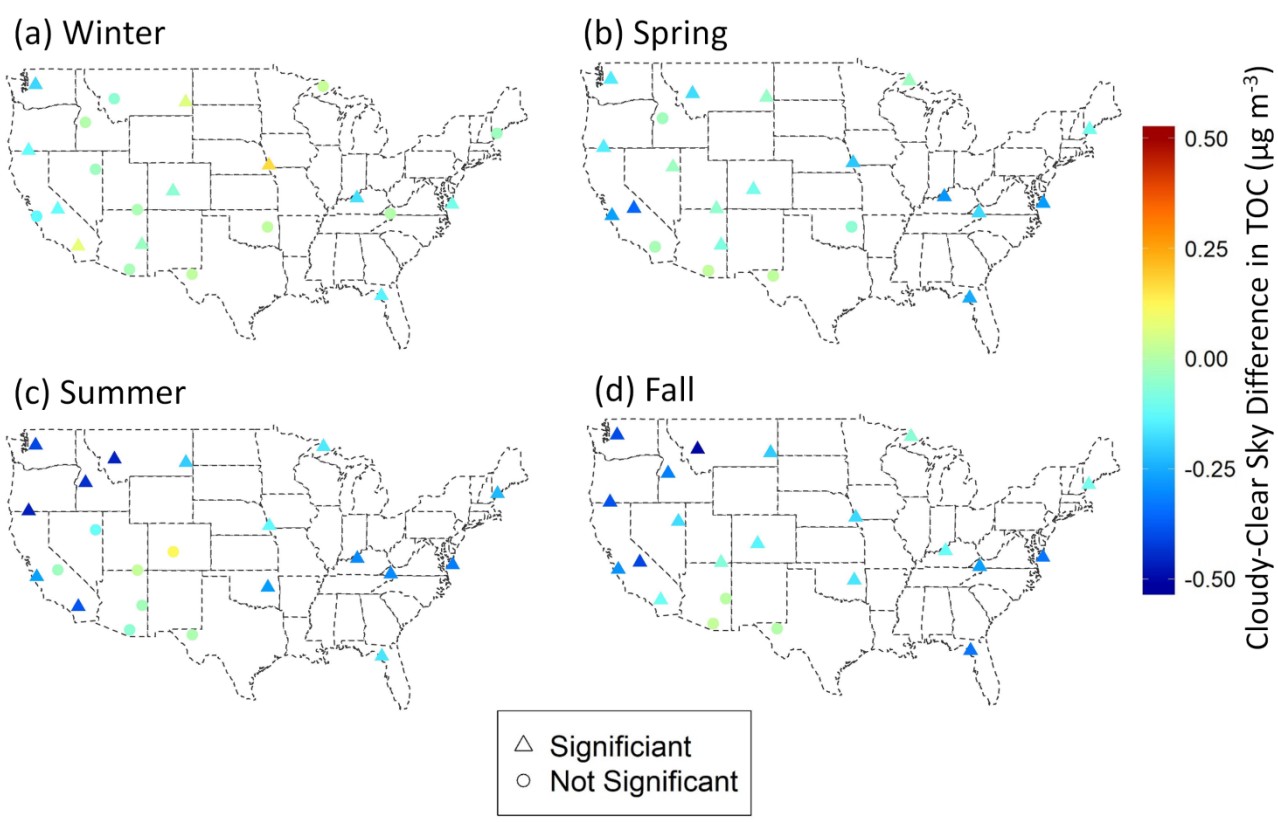

**Figure 7: Maps of the difference in TOC mass concentration medians (Cloudy-Clear Sky) for all regions from 2010-2014 for (a) winter, (b) spring, (c) summer, and (d) fall. The color of the point corresponds to the magnitude of the difference. Triangles indicate that median differences are significant by the Mann-Whitney U Test.**





Figure 8: Box plots of cloudy and clear sky distributions of PM$_{2.5}$ and chemical constituent mass concentrations (µg m$^{-3}$) and RH in the Mid South for each season from 2010-2014. The width of the box plot is proportional to the number of observations. Note that potential outliers are not shown but are used in calculations. Asterisks denote Cloudy and Clear Sky differences that are *not* significant (p<0.05) by the Mann-Whitney U Test.