# Peer review of "Differences in Fine Particle Chemical Composition on Clear and Cloudy Days"

_Atmospheric Chemistry and Physics, 2020_

## Referee Comment (RC1) · Anonymous Referee #1 · 9 Apr 2020

Differences in fine particle chemical composition on clean and cloudy days

This manuscript looks at differences in aerosol data from the IMPROVE network on clear and cloudy days. The difference is important because satellite optical depth measurements only work on clear days and could therefore be biased if aerosol concentrations are systematically different on cloudy days. Since satellites measure aerosol at ambient RH, such a bias could arise both from differences in the amount of dry aerosol and in its water uptake.

The concept of this manuscript is an excellent one – a simple study that fills a gap in the literature and has relevance to satellite measurements. It absolutely should be published in some form. The manuscript, however, could be much, much better if the logical flow were better and the figures were better organized.

Specifics:

Lines 118-119. I disagree with not including the water uptake by organic aerosol. Yes, it is not known as well as for sulfate and nitrate. But ignoring it will give a worse answer than putting in a reasonable value. It is known that the kappa for organics in relevant rural aerosol is often something like 0.2 or 0.3 (as just one example reference Chang et al., 10.5194/acp-10-5047-2010). I looked up the Jathar and Metzger references and they do not say that the OC water uptake is not known well enough to put in a best estimate.

Lines 212-220 (and elsewhere for species other than TOC): This paragraph follows a frequent bias in the literature by talking more about sources than sinks. Aerosol concentrations are also higher during clear sky periods because removal by precipitation is more frequent in cloudy sky periods (e.g. Grandet et al. doi:10.5194/acp-13-3177-2013; and later paper by Gryspeerdt). On the source side mentioning both photochemistry and stagnation events is good. I would also suspect that fire frequency is important for differences between clear and cloudy periods, maybe especially in the eastern US where there is small-scale agricultural burning.

My two big comments are about the figures and the logical ordering of the manuscript. Although it is there in the text if you read really carefully, the overall manuscript doesn't really present in a logical order but instead jumps far too quickly to aerosol liquid water (ALW). I kept wanting to see the differences in concentration shown before the next step of computing ALW. Most of the figures for the concentration differences mysteriously omit organics, one of the most abundant species. Finally, the relevant quantity for comparing to satellites is not ALW. It is the wet aerosol (dry plus liquid water). No existing satellite can measure aerosol water content – so why choose this as the basis for your analysis when the motivation for the entire project is biases in satellite retrievals?

You will have a much better paper if you organize the results by first showing the differences in concentrations, then showing hygroscopicity and how that translates to wet aerosol. A good

example of the poor organization is that Figure 1 is about ALW, which is a derived quantity that uses the information in Figure 2. Any reader just skimming the paper would get confused seeing Figure 1 come first. And even within figures information is poorly organized. In Figure 1, why isn't "mixed" in between clear and cloudy? In Figure 2, why is RH in between PM2.5 and its constituents?

In summary, start with the concentration differences as measured by IMPROVE, then extend those differences to ambient RH, as measured by satellite retrievals.

Another part of the paper that really needs work is the figures. Some figures simultaneously have too many panels and don't convey enough information. Figures 3 to 7 are almost illegible.

If it stays, Figure 1 could be done as one panel with grouped bars (just to be clear what I mean, googling "grouped bar chart" will show what I mean – I'm sure you use them all the time). Except you don't really need Figure 1 – it could be combined as a last column in Figure 2, more like Figure 8.

Figure 2 could be done in 4 panels, not 16. Except it should be 6 panels – it should include TOC and wet aerosol. The columns should be organized logically One good way to organize would be left-to-right to show SO4, NO3, and TOC, then dry PM2.5 (and label it "dry" for clarity), then RH, then wet aerosol.  Figure 8 should follow FIgure 2.

Figures 3 to 7 are extremely hard to read, even at high magnification on the screen. And the color coding is a poor choice for quantitative information. There is research showing that bar and line graphs are read more accurately than are color codes or pie graphs. Readers can correctly discern quantitative changes in bar and line graphs that they can't accurately judge in other formats. Contour plots are good, too. Also, using five repetitive figures (3 to 7) doesn't really work very well. I'd try very, very hard to make some sort of bar graphs that are tied to a map and to put more than one species on the same plot. For ideas look at the way the IMPROVE data were plotted with bar graphs tied to a map in Hand et al. 2012 (doi:10.1029/2011JD017122) or line graphs superimposed on a map in Murphy et al. 2008 (www.atmos-chem-phys.net/8/2729/2008/).

I can't say I can see exactly how you should do the map plots. Please understand as a reviewer one sees figures that work well and figures that don't work. Your figures 3 to 7 don't work well.

There is distracting use of color: for there is no reason why the column headings in Figure 2 should be in colored fonts, and no particular reason why the "clear" should be brownish and the "cloud" should be blue.

[Figure]

**Figure 2a.** IMPROVE 2005–2008 regional monthly mean PM$_{2.5}$ mass concentrations ($\mu$g m$^{-3}$) surrounded by PM$_{2.5}$ reconstructed fine mass fractions for the eastern United States, including the Virgin Islands region. The letters on the *x* axis correspond to the month, and "A" corresponds to "annual" mean. Ammonium sulfate (AS) in yellow, ammonium nitrate (AN) in red, particulate organic matter (POM) in green, light-absorbing carbon (LAC) in black, soil in brown, and sea salt in blue. The shaded area corresponds to the regions that comprise the sites used in the analysis, shown as dots.

Including this here isn't about the content in this figure from Hand et al. Instead, I'm putting this in as an example of how you might try to plot your data to be more legible than Figures 3 to 7 in the manuscript. Instead of the bar plots of composition by month you could have clear/cloud data for major constituents or something like that.

---

## Referee Comment (RC2) · Anonymous Referee #2 · 22 May 2020

This manuscript takes a look at some of the potential biases that arise when satellites are used to validate/inform the modeling/interpretation of ground aerosol measurements (specifically for the continental US and the MODIS satellite sensors). Valid aerosol satellite retrievals for scenes impacted by clouds are challenging, even in the free troposphere (e.g. above the cloud deck) and when looking specifically at ground data, they basically result in a large fraction of measurements being discarded. Christiansen et al ask the question how this missing data impacts aerosol composition and one related quantity, aerosol liquid water (ALW), using IMPROVE data for the former and ISORROPIA runs to estimate the later. They conclude that there is indeed a significant bias in both quantities across most seasons in the US.

Since clear-sky AOD (which is impacted by both dry aerosol concentration and ALW) is one of the preferred ways to evaluate GCM performance over large domains, the bias described in this paper is very relevant to the ACP readership. The chosen approach is simple, elegant and reasonably robust. While it has been used before to explore the PM2.5 bias in the CONUS (Christopher and Gupta, 2010), this manuscript improves on the method, uses more recent data and more advanced MODIS products and adds seasonality and ALW to the mix.  So it will certainly be a useful addition to the literature.  Not unlike Reviewer #1, however, I found some aspects of the logical flow of the current manuscript confusing, so here are some suggestions that hopefully will improve the presentation of what is overall very nice work.

Major comments:

- There are two interrelated questions that the manuscript is currently trying to addresses:
    a) Does the presence of boundary layer clouds impact aerosol composition on the ground in rural US locations and if so, how?
    b) Does the aerosol in cloudy satellite scenes look different than clear sky scenes and soes this result in a systematic bias for AOD analysis that only use clear sky scenes?
  While most of the manuscript focuses on a), b) is repeatedly mentioned in the abstract and conclusions, although never really directly addressed. Now, if (a) is indeed the main objective, the current approach is a little bit less than ideal. To wit:
    o If I follow correctly, the cloudmask used in this manuscript does NOT differentiate between low and high clouds (although this information is provided by MODIS). If the goal of this paper is (b), that is a valid choice. But if the goal is to actually explore how boundary layer clouds affect chemistry, this should be certainly changed, and the analysis repeated with low clouds exclusively. I would expect that this will actually strengthen the trends observed and won't be statistically too costly, especially in the Western US.
    o While relating the cloudmask to the most often used AOD sensor is certainly very useful to explore (b), it results in a pretty clear bias since AQUA and TERRA overpasses over the CONUS are only once a day each, and none of them after 2 pm (this info is missing currently from the manuscript and needs to be added for context), while the compositional data is taken for a 24 h period. Especially for the late Spring and Summer, where both the maximum cloud and photochemistry activity is later in the day, this will

lead to some misclassification. So again, if the impact of BL cloud chemistry is the focus, using data from a geostationary satellite (e.g GOES16 or Himawari-8, both of which have similar pixel sizes to MODIS) for full day coverage would make more a lot more sense and result in likely more robust results that using MODIS.

On the other hand, if (b) is the main objective, as Reviewer #1 mentioned in his review, the current manuscript does not really try to relate the changes in PM2.5 and ALW to an actual bias in the MODIS product.

From my reading of the manuscript, the goal seems to be to answer (a) and use (b) for illustrative purposes, which is a good choice in my opinion, especially since

- the authors have already published a manuscript with a similar approach where they look in more detail at the AOD impacts (Christiansen et al, 2019) that could be mentioned in some more length in the conclusions, and
- as discussed both in the manuscript and below, there is just too much uncertainty in the IMPROVE data to attempt closure, especially since FT aerosols are not included and likely play a role (again as discussed in Christiansen et al, 2019)

But in that case, I would certainly ask the authors to please clarify and illustrate the trade-offs in their choices of satellite products. Switching to geostationary data is probably outside the scope of this project, but the option should be discussed (and maybe considered for future analysis of similar dataset).

- Related to the previous point, the current manuscript spends considerable space talking about cloud affecting the FT aerosol, which is not really the focus of the paper (e.g. not clear to me why the fact that most aerosol measurements from aircraft exclude the cloud itself is relevant to this work, which deals with the aerosol below), so I would consider shortening those sections (mostly the intro) significantly to avoid confusion and for better flow. It also never really mentions the most important factor that matters for what the authors are exploring here, aerosol ground concentrations, namely clouds blocking any retrieval below cloud height. It is true that the MODIS team has done a great job trying to retrieve AOD ABOVE cloudy scenes, as the manuscript mentions, but below is just not feasible if it's too cloudy, one of the reasons why the author's analysis is so valuable: it's not faulty data, but missing data that they are addressing.

- I would strongly suggest to move the PM2.5 analysis at the beginning of the discussion, since it is the simpler metric to start with, and then go into the detailed compositional/ALW analysis. Most importantly, since this type of analysis has been done before, both in Christopher and Gupta with older MODIS data product and , for non-cloudy data using the new algorithm in Chudonowski et al, 2013, it would actually allow for a direct comparison of the observed PM2.5 with previous studies, which would significantly strengthen the analysis. It would also, to the extent possible, be useful if the authors could comment on why Christopher and Gupta came to the opposite conclusion.

- The speciated IMPOVE data is certainly not ideal for calculating ALW. The authors have done one sensitivity analysis to investigate the impact of dust. However, a similar analysis for the missing ammonium and OA is missing.

- The impact of ammonium, since it will be associated with nitrate and sulfate, will, in the absence of good acidity measurements, just be a positive offset on the current trends,

as the authors write. Still, since for neutralized aerosol it is going to add significant mass and ALW to the analysis, it would be nice to add this explicitly to Fig S2

o I agree with Reviewer #1 that not estimating OA ALW is an odd choice, since OA has a clear seasonal trend that will in some cases make the observed trends likely stronger and there are good published estimates for kappa for oxidized aerosol over the continental US (e.g. Brock et al, 2016 and Shingler et al, 2016). Obviously fresh POA is an issue, but since the analysis is restricted to rural sites there should be very little urban POA. Fire POA if needed can be just filtered by either the MODIS fire flag or some simpler compositional metric (e.g. f(OA)>0.8). So I would suggest that – at least as an additional sensitivity study - the authors add OA ALW, using an OA/OC of 2.1 (used for OOA in newer GCMs like Geos-Chem and CESM2) outside of fire days and a kappa value of ~0.1 (for OA). This would be very close to the assumptions that a modeling study would do and hence give a good sense of the possible effect

- I agree with Reviewer #1 that changing the format of Fig 3 to 7 (and Fig S4) to a single map with barplots for the four seasons (or some variation of this general idea) would make it much easier to take in the information conveyed in those graphs quickly, and would highlight the seasonal trends much better than the current format.

- The effect of wet deposition is likely significant and I would not expect it to be countered by evaporation (at least outside of cloud detrainment). Its sign is opposite to the trends shown in this paper and hence it does not affect the significance of the conclusions. Still, since the authors have the reanalysis meteorology, a simple sensitivity study looking at the differences between cloudy vs cloudy + rainy days would give the reader a better sense of how large the problem is.

Minor comments:

- Line 24:  For OA and sulfate, Ervens et al 2018 have recently looked at FT cloud chemistry as well, please add.

- Line 24-25: Clouds do not redistribute aerosols and gases, convection (or convective clouds, if you will) does. Please rephrase.

- Line 40: This is wrong as written. RH dependent lab studies have been done since the '60s! I am assuming the authors are referring specifically to environmental chamber work exploring aerosol chemistry, where it is true that many older studies were done under dry conditions, but that has changen in the past decade, see e.g. Petters et al, 2017 or Schwantes et al 2019 for recent examples. So please revise accordingly.

- Line 97: It is not clear to me how 17x500m = 50 km, please explain. It would also help to mention some version of this is standard practice for most satellite/ground comparisons and that e.g. Christopher and Gupta, 2010 used 5x5 grids with a resolution of 4 km.

- In that context, it would be worth citing Twohy et al, 2009 that the effect of clouds on AOD extends at least 20 km farther than the actual cloud, and that using the larger detection area will hence certainly improve the accurate detection of *cloud free scenes*

- Line 215: Biogenic emissions have a strong dependence on ambient/soil temperature. Cloudy scenes are likely colder, so in the absence of data this effect could be either sun or temperature

- Line 217: While there might be some uncertainty in OA ALW (mostly related to the O/C ratio of OA) for the inorganic concentrations at this rural locations the overall effect is not that large. So while this statement might be true in other domains, I would revise it for the CONUS.

- Line 240: Jefferson et al show f(RH) that are significantly higher than the values reported in Table 1. Again, it would increase confidence in the analysis if it could be shown that adding ammonium and OC does result in roughly similar growth rates.

References:

Christiansen, A. E., Ghate, V. P. and Carlton, A. G.: Aerosol Optical Thickness: Organic Composition, Associated Particle Water, and Aloft Extinction, ACS Earth Sp. Chem., 3(3), 403–412, doi:10.1021/acsearthspacechem.8b00163, 2019.

Chudnovsky, A., Tang, C., Lyapustin, A., Wang, Y., Schwartz, J. and Koutrakis, P.: A critical assessment of high-resolution aerosol optical depth retrievals for fine particulate matter predictions, Atmos. Chem. Phys., 13(21), 10907–10917, doi:10.5194/acp-13-10907-2013, 2013.

Ervens, B., Sorooshian, A., Aldhaif, A. M., Shingler, T., Crosbie, E., Ziemba, L., Campuzano-Jost, P., Jimenez, J. L. and Wisthaler, A.: Is there an aerosol signature of chemical cloud processing?, Atmos. Chem. Phys., 18(21), doi:10.5194/acp-18-16099-2018, 2018.

Liao, J., Froyd, K. D., Murphy, D. M., Keutsch, F. N., Yu, G., Wennberg, P. O., St. Clair, J. M., Crounse, J. D., Wisthaler, A., Mikoviny, T., Jimenez, J. L., Campuzano-Jost, P., Day, D. A., Hu, W., Ryerson, T. B., Pollack, I. B., Peischl, J., Anderson, B. E., Ziemba, L. D., Blake, D. R., Meinardi, S. and Diskin, G.: Airborne measurements of organosulfates over the continental U.S., J. Geophys. Res. Atmos., 120(7), 2990–3005, doi:10.1002/2014JD022378, 2015.

Petters, S. S., Pagonis, D., Cla, M. S., Levin, E. J. T., Petters, M. D., Ziemann, P. J., Kreidenweis, S. M., Claflin, M. S., Levin, E. J. T., Petters, M. D., Ziemann, P. J. and Kreidenweis, S. M.: Hygroscopicity of Organic Compounds as a Function of Carbon Chain Length and Carboxyl, Hydroperoxy, and Carbonyl Functional Groups, J. Phys. Chem. A, 121(27), 5164–5174, doi:10.1021/acs.jpca.7b04114, 2017.

Schwantes, R. H., Charan, S. M., Bates, K. H., Huang, Y., Nguyen, T. B., Mai, H., Kong, W., Flagan, R. C. and Seinfeld, J. H.: Low-volatility compounds contribute significantly to isoprene secondary organic aerosol (SOA) under high-$NO_x$ conditions, Atmos. Chem. Phys., 19(11), 7255–7278, doi:10.5194/acp-19-7255-2019, 2019.

Shingler, T., Sorooshian, A., Ortega, A., Crosbie, E., Wonaschütz, A., Perring, A. E., Beyersdorf, A., Ziemba, L., Jimenez, J. L., Campuzano-Jost, P., Mikoviny, T., Wisthaler, A. and Russell, L. M.: Ambient observations of hygroscopic growth factor and f (RH) below 1: Case studies from surface and airborne measurements, J. Geophys. Res. Atmos., 121(22), 13,661-13,677, doi:10.1002/2016JD025471, 2016.

Twohy, C. H., Coakley, J. A. and Tahnk, W. R.: Effect of changes in relative humidity on aerosol scattering near clouds, J. Geophys. Res. Atmos., 114(5), 1–12, doi:10.1029/2008JD010991, 2009.

---

## Author Comment (AC1) · 3 Aug 2020

**Reviewer 1**

This manuscript looks at differences in aerosol data from the IMPROVE network on clear and cloudy days. The difference is important because satellite optical depth measurements only work on clear days and could therefore be biased if aerosol concentrations are systematically different on cloudy days. Since satellites measure aerosol at ambient RH, such a bias could arise both from differences in the amount of dry aerosol and in its water uptake.

The concept of this manuscript is an excellent one – a simple study that fills a gap in the literature and has relevance to satellite measurements. It absolutely should be published in some form. The manuscript, however, could be much, much better if the logical flow were better and the figures were better organized.

We thank the Reviewer for their comments. We respond to specific comments below and indicate changed text in red.

Lines 118-119. I disagree with not including the water uptake by organic aerosol. Yes, it is not known as well as for sulfate and nitrate. But ignoring it will give a worse answer than putting in a reasonable value. It is known that the kappa for organics in relevant rural aerosol is often something like 0.2 or 0.3 (as just one example reference Chang et al., 10.5194/acp-10-5047-2010). I looked up the Jathar and Metzger references and they do not say that the OC water uptake is not known well enough to put in a best estimate.

The Reviewer is correct that the Jathar reference does not say that organic water uptake is too uncertain to estimate. The Reviewer is also correct that organic species can dramatically modulate particle hygroscopicity. A sensitivity investigating organic ALW concentrations performed by Nguyen et al. (2015) specifically for IMPROVE sites shows that OC contributions to aerosol water impacts absolute values in mass concentrations, but not interpretation of overall trends when compared to ALW calculated from inorganic components. We also estimated [ALW], again specifically at IMPROVE sites, with OC mass concentrations using the individual fractions (Christiansen et al., 2019) in an unsuccessful attempt to better connect to satellite-derived AOD. We were unable to broadly improve statistical relationships between AOD and surface [ALW] through inclusion of OC, and this made us cautious about considering organic species here. As suggested by the Reviewer, we re-did all ISORROPIA calculations for all sites over the entire time period and employ a kappa value of 0.3 to estimate ALW from organic components. Our main conclusion that finds ALW mass concentrations are larger during cloudy times than clear sky times across the CONUS (see new figure below, now Figure 4) remains true and consistent with the observed trend in inorganic ALW only. We include this information in the manuscript in both the methods and results sections and have a new figure explicitly estimating inorganic and organic contribution to ALW. We also more accurately reflect the cited studies:

Organic hygroscopicity values are uncertain (Metzger et al., 2018; Nguyen et al., 2015) and the magnitude of water uptake by organics varies by location (Jathar et al., 2016)...

And change the text to discuss ALW due to organic compounds.

Methods:

Organic hygroscopicity values are uncertain (Metzger et al., 2018; Nguyen et al., 2015) and the magnitude of water uptake by organics varies by location (Jathar et al., 2016), in particular at IMPROVE sites (Christiansen et al., 2019). We provide an estimate of organic ALW using a relevant hygroscopicity value for rural aerosol of 0.3 (Chang et al., 2010; Nguyen et al., 2014). Organic speciation at IMPROVE locations changes in time and space (Christiansen et al., 2020) and the suitability of applying a constant value for organic hygroscopicity is difficult to quantitatively assess. Organic ALW is estimated as in Christiansen et al. (2019) and Nguyen et al. (2015). Briefly, we use κ-Kohler theory and the Zdanovskii-Stokes-Robinson (ZSR) mixing rule (Eq. 1).

$$V_{w,o} = V_o \kappa_{org} \frac{a_w}{1 - a_w} \tag{1}$$

Here, the water activity  $(a_w)$  is assumed to be equivalent to RH,  $V_o$  and  $V_{w,o}$  are the volumes of organic matter and water from organic species, respectively, and  $\kappa_{org}$  is the organic hygroscopicity parameter.  $V_o$  is determined by dividing organic mass (OM) by 1.4 g cm-3 (Christiansen et al., 2019). OM is calculated from IMPROVE-measured OC with site- and time-specific OM:OC ratios, which are estimated via a mass balance method, as described in Malm et al. (2020) and Christiansen et al. (2020). Temperature and RH values ...

To the Results we add text in addition to the new figure 4:

Mass concentrations of TOC are nearly always higher during Clear Sky times than Cloudy (Fig. 4, Table S7) in all chemical climatology regions across the CONUS, with the largest differences during summer and fall. The patterns are unique and consistent with SOA. Summertime wildland fires in the west and prescribed burning during spring and fall in the east may obscure interpretation due to large episodic primary OC emissions (Spracklen et al., 2007; Tian et al., 2009; Zeng et al., 2008). However, at IMPROVE monitoring locations, secondary organic aerosol (SOA) contribution to TOC dominates over contribution from primary sources (Carlton et al., 2018a). The most pronounced differences in Clear Sky and Cloudy TOC occur in summer in regions where precursor biogenic VOC emissions that form SOA are substantial (Donahue et al., 2009; Gentner et

al., 2017; Youn et al., 2013). Further, increased sunlight and higher temperatures under Clear Sky conditions (Table S6) lead to higher biogenic VOC emissions that form SOA (Laothawornkitkul et al., 2009; Sakulyanontvittaya et al., 2008) and enhanced photolysis rates that facilitate hydroxyl radical production important to SOA formation (Tang et al., 2003). These findings suggest differing organic chemical composition in TOC, on Clear Sky and Cloudy days.

Cloudy period ALW mass concentrations are higher than Clear Sky in all seasons from both inorganic and organic contributions, with few exceptions (Fig. 3, Table S8). The largest Cloudy and Clear Sky ALW differences are observed in the central and eastern US during winter. The pattern in higher ALW during Cloudy periods is opposite the pattern of dry PM2.5 mass and arises from a combination of higher RH and changing aerosol composition that affects hygroscopicity. Nitrate is the most hygroscopic species considered in this analysis and high Cloudy NO3- mass concentrations increase particle hygroscopicity to facilitate ALW during these times, despite lower overall dry PM2.5 mass. Clear Sky and Cloudy changes in the precise chemical composition of organic compounds and their impacts on ALW remain critical open questions.

Fig. 4. Distributions of a) inorganic ALW, b) organic ALW, and c) total (inorganic + organic) ALW during Clear Sky (yellow) and Cloudy (blue) times in all seasons across the CONUS. The width of the boxplot is proportional to the amount of observations that make up each distribution. Note that potential outliers are not shown, but are used in calculations.

Lines 212-220 (and elsewhere for species other than TOC): This paragraph follows a frequent bias in the literature by talking more about sources than sinks. Aerosol concentrations are also higher during clear sky periods because removal by precipitation is more frequent in cloudy sky periods (e.g. Grandet et al. doi:10.5194/acp-13-3177-2013; and later paper by Gryspeerdt). On the source side mentioning both photochemistry and stagnation events is good. I would also suspect that fire frequency is important for differences between clear and cloudy periods, maybe especially in the eastern US where there is small-scale agricultural burning.

We completely agree and the Reviewer is correct that we have not sufficiently discussed sinks. It is important to also note that the cloud definition we used employs the whole column (e.g., includes high level cirrus clouds) and most cloud droplets do not precipitate. However, we recognize that precipitation is an important factor in aerosol removal and certainly contributes to the observed differences. We also hypothesize that aerosol with more liquid water are physically larger and therefore, during cloud times (higher ALW) likely to dry deposit more quickly as well. We agree with the Reviewer there is a bias toward production. Our changed text:

**Methods:**

The impacts of wet deposition due to precipitation and dry deposition (i.e., particles are physically larger and more likely to deposit when water uptake is higher (Carlton et al., 2020)) are unconstrained in this analysis.

**Results:**

There is an increased likelihood of aerosol removal due to scavenging by precipitation during Cloudy times, and this may contribute to differences in mass concentrations. However, the cloud definition employed here uses the entire column (i.e., non-precipitating cirrus and stratus clouds are included), and the majority of cloud droplets evaporate (Pruppacher and Klett, 2010).

In the case of fires, the Reviewer is correct that fire frequency is an important factor for TOC concentrations. We have included a statement that acknowledges the role of fires in OC concentrations. Fires most likely obscure TOC Clear Sky/Cloudy differences, as the amount of OC produced by fires is much larger than differences in Clear Sky and Cloudy TOC concentrations in the absence of fires. We amend the text to better include recognition of fire:

Summertime wildland fires in the west and prescribed burning during spring and fall in the east may obscure interpretation due to large episodic primary OC emissions (Spracklen et al., 2007; Tian et al., 2009; Zeng et al., 2008). However, at IMPROVE monitoring locations, secondary organic aerosol (SOA) contribution to TOC dominates over contribution from primary sources (Carlton et al., 2018a). The most pronounced differences in Clear Sky and Cloudy TOC occur in summer in regions where precursor biogenic VOC emissions that form SOA are substantial (Donahue et al., 2009; Gentner et al., 2017; Youn et al., 2013). Further, increased sunlight and higher temperatures under Clear Sky conditions (Table S6) lead to higher biogenic VOC emissions that form SOA (Laothawornkitkul et al., 2009; Sakulyanontvittaya et al., 2008) and enhanced photolysis rates that facilitate hydroxyl radical production important to SOA formation (Tang et al., 2003). These findings suggest differing organic chemical composition in TOC , on Clear Sky and Cloudy days.

My two big comments are about the figures and the logical ordering of the manuscript. Although it is there in the text if you read really carefully, the overall manuscript doesn't really present in a logical order but instead jumps far too quickly to aerosol liquid water (ALW). I kept wanting to see the differences in concentration shown before the next step of computing ALW. Most of the figures for the concentration differences mysteriously omit organics, one of the most abundant species. Finally, the relevant quantity for comparing to satellites is not ALW. It is the wet aerosol (dry plus liquid water). No existing satellite can measure aerosol water content – so why choose this as the basis for your analysis when the motivation for the entire project is biases in satellite retrievals?

The Reviewer is correct that total organic carbon (TOC) should be included in these figures. TOC was not the focus of this paper, and we previously excluded it in our ALW estimates. However, TOC is included in the total PM2.5 concentration and should be included in our figures. We now include TOC explicitly in the text and Figures. For example, TOC text is described above. Several Figures in the main text, appendix and supplemental information include TOC: